# Three Cases of Lymphocytic Infiltration of the Eyelid

**Kyoko Sugioka** [1], **Akinobu Hayashi** [2], **Masako Ichishi** [2], **Yasuko Sugimoto** [3], **Koji Habe** [1] **and Keiichi Yamanaka** [1,*]

1   Department of Dermatology, Graduate School of Medicine, Mie University, Tsu, Mie 514-8507, Japan; kyoko.ikeda.725@gmail.com (K.S.); habe-k@clin.medic.mie-u.ac.jp (K.H.)
2   Department of Oncologic Pathology, Graduate School of Medicine, Mie University, Tsu, Mie 514-8507, Japan; ahayashi@doc.medic.mie-u.ac.jp (A.H.); masako-i@doc.medic.mie-u.ac.jp (M.I.)
3   Department of Dermatology, Matsusaka City Hospital, Matsusaka, Mie 515-8544, Japan; yakosugi@hotmail.com
*   Correspondence: yamake@med.mie-u.ac.jp; Tel.: +81-59-231-5025; Fax: +81-59-231-5206

**Abstract:** Lymphocytic infiltration of the skin (LIS), first reported by Jessner and Kanof in 1953, is a disease of unknown etiology characterized by erythematous papules and plaques on the head, neck, and upper back and histopathological findings of a normal epidermis with underlying lymphocytic infiltration of the reticular dermis without mucin deposition. A 69-year-old man and a 21-year-old woman presented with edematous indurative erythema of the left upper eyelid. Lymphocytic infiltration of the dermis with CD4+ T cell predominance was noted on biopsy. A 68-year-old man presented with a four-year history of recurrent edematous indurative erythema of the right upper eyelid that extended up to the right cheek. Predominantly dermal infiltration of CD8+ T lymphocytes was found on biopsy. We treated all three patients with 8–16 mg of methylprednisolone daily, and the erythema and induration improved. CD4+ T cells were predominant in the acute phase (patients 1 and 2), whereas CD8+ T cells were predominant in the chronic phase (patient 3). CD8+ T cells may be involved in LIS recurrence. Lymphocytic infiltration of the eyelid may be associated with isolated circumscribed, edematous, indurative, colorless lesions that are responsive to daily low-to-middle doses of oral methylprednisolone.

**Keywords:** lymphocytic infiltration of the skin; methylprednisolone; CD4+ T cell; CD8+ T cell; lymphocytic infiltration of the eyelid

## 1. Introduction

Lymphocytic infiltration of the skin (LIS) is a rare dermatological condition of unknown etiology that was first reported by Jessner and Kanof in 1953 [1]. It is characterized by erythema, papules, and plaque-like eruptions that are mainly located on the face, neck, or upper back. The lesions are usually painless and non-pruritic. The presentation of LIS can vary, and remission and exacerbation may occur over months or years. The differential diagnoses of LIS include pseudolymphoma, sarcoidosis, and lupus erythematosus tumidus. No definitive therapy has been developed; intermittent administration of topical steroids can be useful, but it is not effective in many patients [2]. Here, we report the cases of three patients with lymphocytic infiltration of the eyelid (LIE) who presented with single, edematous, colorless, erythematous, indurative plaques on the eyelid. One of them also presented with infiltrated erythema of the cheek.

## 2. Case Report

We show the histopathological and immune histopathological findings of the skin biopsy from 3 cases in Figure 1.

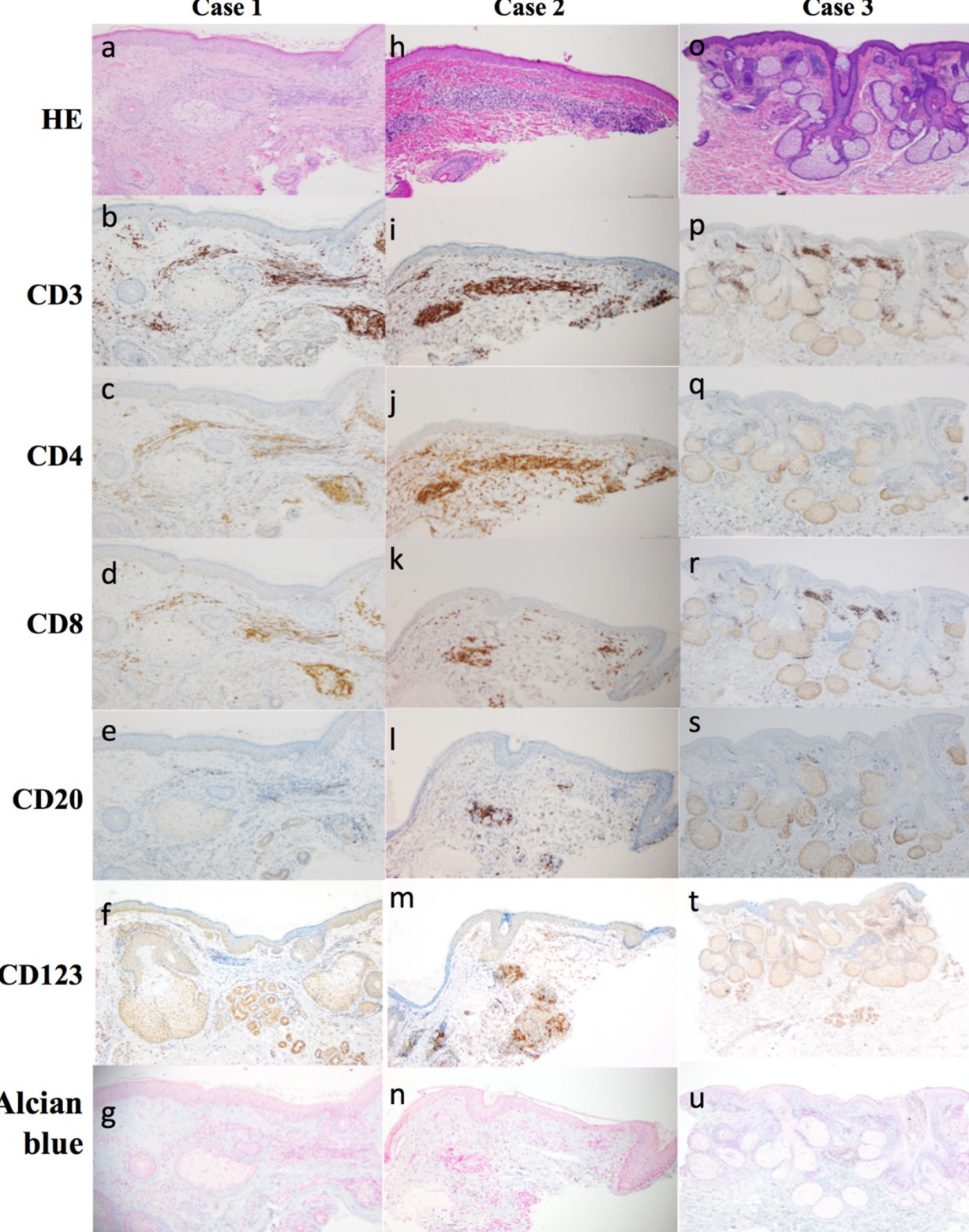

**Figure 1.** (**a**,**h**,**o**) Lymphocytic infiltration of the reticular dermis (hematoxylin and eosin, ×100). (**b**,**i**,**p**) Infiltration of CD3+ T cells (CD3 immunostaining, ×100). (**c**,**j**,**q**) Infiltration of CD4+ T cells (CD4 immunostaining, ×100). (**d**,**k**,**r**) Infiltration of CD8+ T cells (CD8 immunostaining, ×40). (**e**,**l**,**s**) Infiltration of CD20+ B cells (CD20 immunostaining, ×100). (**f**,**m**,**t**) Infiltration of CD123+ cells (CD123 immunostaining, ×100). (**g**,**n**,**u**) Alcian blue staining (×100). CD4+ T cell predominance was observed in patients 1 and 2, but not in patient 3, who showed CD8+ T cell predominance. Few CD20+ B cells were observed in all three patients. CD123 + suggestive for plasmacytoid DC were negative in cases 1 and 3, and partially positive in case 2. Mucin deposition was not detected. HE: hematoxylin and eosin.

## 2.1. Case 1

A 69-year-old man presented to our clinic with a four-month history of edematous erythema of the left upper eyelid. He had a history of hypertension and polymyalgia rheumatica. His antinuclear antibody titer was 1:40, with a homogeneous pattern. His serum angiotensin-converting enzyme (7.2 U/L), immunoglobulin (Ig) G (1290 mg/dL), IgG4 (65 mg/dL), creatine phosphokinase (114 U/L) levels, and C-reactive protein (CRP) (0.03 mg/dL) were within normal limits. Additionally, his white blood cell (WBC), including absolute neutrophil and lymphocyte counts was within normal (WBC: 3430/μL, neutrophil: 1400/μL, lymphocyte: 1640/μL). Physical examination revealed edematous erythema with left upper eyelid infiltration (Figure 2a). Skin biopsy of the edematous, erythematous eyelid lesion showed a normal epidermis with dense lymphocytic infiltration in the reticular dermis (Figure 1a). Immunohistochemistry showed a predominance of CD4+ T cells compared to CD8+ T cells and few CD20+ B cells (Figure 1b–e). Mucin deposition was not detected on Alcian blue staining (Figure 1f). He was treated with 16 mg of oral methylprednisolone daily. Since we observed a good response (Figure 2b), we tapered the methylprednisolone dose by 4 mg every four weeks until the dose was 6 mg/day. This dose was further tapered over eight months, and the total duration of treatment was 11 months (Figure 2c). Throughout the clinical course, all laboratory data was within the normal range.

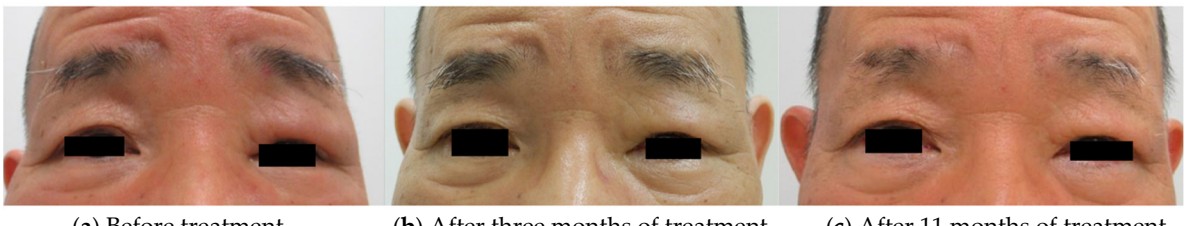

(**a**) Before treatment    (**b**) After three months of treatment    (**c**) After 11 months of treatment

**Figure 2.** (**a**,**b**) The patient's left upper eyelid is swollen, and a single, erythematous, indurated lesion is seen. (**c**) Improvement in swelling and erythema after treatment.

## 2.2. Case 2

A 21-year-old woman presented to our clinic with a three-month history of edematous induration of the left upper eyelid. She had no history of any other major disease. Her serum IgG4 (16.2 mg/dL), C1-inhibitor (81%), and C4 levels (20 mg/dL) were within normal limits. CRP was mildly elevated (0.78 mg/dL). Additionally, her WBC, including absolute neutrophil and lymphocyte counts was also within normal (WBC: 3800/μL, neutrophil: 1748/μL, lymphocyte: 1089/μL). On physical examination, we observed erythematous swelling of the left upper eyelid (Figure 3a). Skin biopsy of the left eyelid showed lymphocytic infiltration of the dermis (Figure 1h). Immunohistochemistry showed a predominance of T cells, a majority of which were CD4+ cells. A few CD20+ B cells were also observed (Figure 1j–l). There was no mucin deposition in the dermis (Figure 1n). We treated her with 16 mg of oral methylprednisolone daily. Since we observed a good response (Figure 3b), we tapered the methylprednisolone dose by 4 mg every eight weeks until the dose was 8 mg/day, following which we further reduced it by 2 mg every four weeks. The total duration of treatment was 11 months (Figure 3c). Throughout the clinical course, all laboratory data was within the normal range.

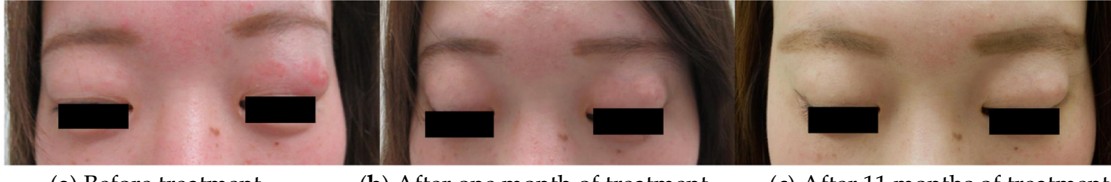

(**a**) Before treatment  (**b**) After one month of treatment  (**c**) After 11 months of treatment

**Figure 3.** (**a**,**b**) The patient's left eyelid shows erythematous swelling. (**c**) Improvement in swelling and erythema after treatment.

*2.3. Case 3*

A 68-year-old man presented with a four-year history of erythematous plaques on the right eyelid. He experienced multiple episodes of symptom remission and recurrence over a period of four years. We observed edematous erythema on his right cheek (Figure 4a). He had no fever, local warmth, or cheek pain. He had a history of rheumatoid arthritis. His serum IgG (983 mg/dL), IgG4 (32 mg/dL), CRP (0.03 mg/dL), and WBC, including absolute neutrophil and lymphocyte counts, were within normal limits (WBC: 4460/μL, neutrophil: 1953/μL, lymphocyte: 1971/μL). He tested negative for the rheumatoid factor. The differential diagnoses were sarcoidosis and granuloma faciale. Skin biopsy revealed a normal epidermis with dense lymphocytic infiltration of the reticular dermis (Figure 1o). There was no mucin deposition in the dermis. Immunohistochemical staining revealed aggregates of CD3+ and CD8+ T cells, but not of CD4+ T cells, in the dermis (Figure 1p–r). A few CD20+ B cells were observed (Figure 1s). The diagnosis of LIE was made, and we started treatment with 8 mg/day of oral methylprednisolone. Due to the good response (Figure 4b), we tapered the methylprednisolone dose by 2 mg every two months. The duration of treatment was eight months (Figure 4c). Throughout the clinical course, all laboratory data was within the normal range.

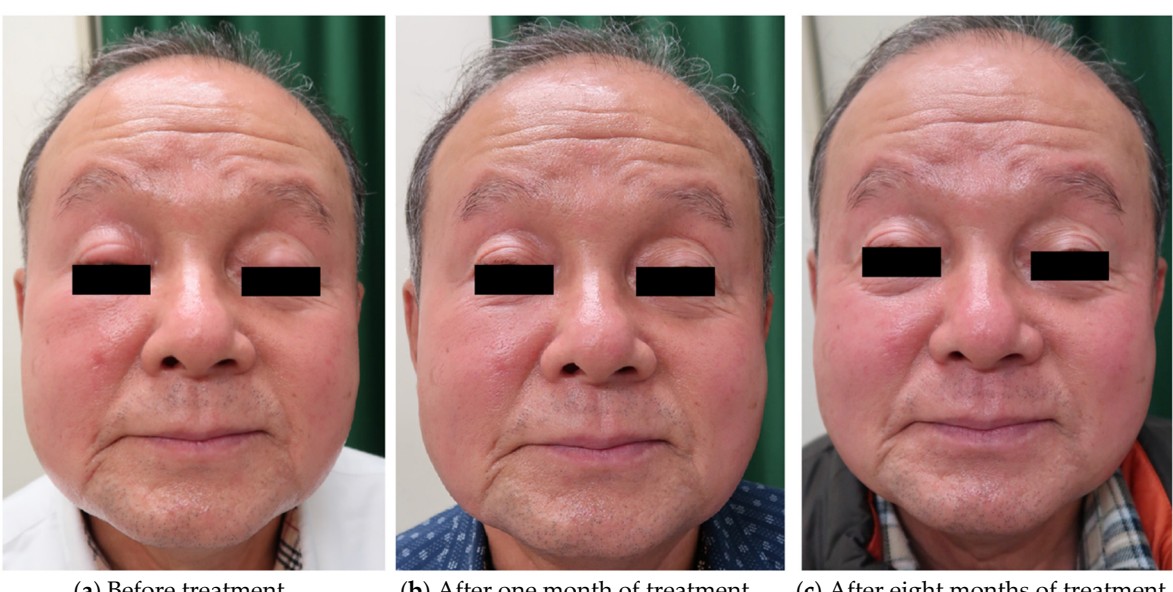

(**a**) Before treatment  (**b**) After one month of treatment  (**c**) After eight months of treatment

**Figure 4.** The patient's right upper eyelid shows a single, erythematous, indurated lesion. His right cheek also shows edematous erythema.

## 3. Discussion

LIS is a rare dermatological condition that was first reported by Jessner and Kanof in 1953 [1]. It is often difficult to diagnose. The clinical differential diagnoses in cases 1 and 2 were pseudolymphoma, sarcoidosis, lupus erythematosus tumidus, and Morbihan disease, whereas those in case 3 were sarcoidosis, granuloma faciale, and Morbihan disease. All

three patients showed histopathological findings of a normal epidermis and lymphocytic infiltration of the upper dermis without mucin deposition. Pseudolymphoma is associated with histopathological findings of clusters of lymphocytes without nuclear atypia that form lymph follicle-like structures in the dermis. Patients with sarcoidosis show epithelioid cell granulomas; those with lupus erythematosus tumidus show atrophy, follicular plugging, mucin deposition, and basal layer vacuolation with a background of photosensitivity; and those with granuloma faciale show granulomatous reactions. Morbihan disease is a rare condition characterized by chronic, persistent, solid, erythematous facial edema [3]. Most patients with Morbihan disease have a history of acne or rosacea complications, and they show non-specific histopathological findings, including dermal edema; blood vessel dilatation; presence of lymphocytes, neutrophils, and perivascular and perifollicular histiocytes; perifollicular fibrosis; and, rarely, an increase in the number of mast cells [4–7]. None of the three patients had a history of acne or rosacea. Furthermore, they did not show histopathological findings of dermal edema, blood vessel dilatation, or infiltration of neutrophils and mast cells. Consequently, they did not meet the diagnostic criteria of any of the abovementioned diseases. Furthermore, all of them had a single, edematous, indurated lesion, which was less erythematous than is usually seen in LIS. Therefore, we diagnosed them with LIE.

While the etiology of LIS remains unclear, the involvement of T cell damage has been suggested [8]. Willemze et al. reported that infiltrates in LIS were mainly composed of CD4+ T cells [8]. On the other hand, Tomasini et al. observed a cluster of plasmacytoid dendritic cells (PDC) in the perivascular and periadnexal regions of the dermis in patients with LIS [9]. PDC activation has been thought to result in CD8+ T cell predominance, and Williams et al. suggested that PDC-directed migration of lymphocytes leads to an inappropriate accumulation of CD8+ T lymphocytes in the dermis, resulting in CD8+ T cell predominance [10]. B cells expressing follicular center differentiation are found close to superficial vessels [11]. The other hand, in our study, the three patients underwent skin biopsy in different phases of the disease course. Skin biopsy was performed in the acute phase in patients 1 and 2, and it showed predominant CD4+ T lymphocyte infiltration. In contrast, skin biopsy was performed in the chronic phase in patient 3, and it showed predominant CD8+ T lymphocyte infiltration. We speculated that infiltration of CD4+ T cells predominates in the acute phase, and infiltration of CD8+ T cells predominates in the chronic and recurrent phases. The distribution of CD4+ and CD8+ T cells can be examined after treatment completion. CD123 is the one of the surface markers of PDC, and CD123 was unstained in cases 1 and 3, and partially positive in case 2. Additionally, few CD20+ B cells were observed in all three patients. Considering above histopathological findings, we may propose the idea LIE is the distinct different category apart from LIS.

There is no effective therapy for LIS and LIE. Reports exist of patients who were treated with topical steroids, but many of them relapsed [2,12]. There are few reports of patients who were treated with systemic steroids. Since our patients' lesion mainly extended from the superficial layer to the middle layer of the dermis, we considered topical therapy to be insufficient and treated them with systemic steroids. The steroid dose was tapered over a period of 11 months, and all three patients were in remission. LIE may be characterized by circumscribed, edematous, indurated, erythematous lesions that respond well to oral methylprednisolone therapy.

## 4. Conclusions

The etiologies of LIS and LIE remain unknown, and no definitive therapy is available. Previous reports have suggested the involvement of CD8+ T cells; however, skin biopsy findings from different stages of LIE showed that CD4+ T cells may predominate in the acute phase and CD8+ T cells may predominate in the chronic phase. Although few such reports exist, we successfully treated three patients with LIE with systemic steroids. Thus, systemic steroids can be effective for the treatment of LIE.

**Author Contributions:** A.H. and M.I. performed immune histochemical staining. K.S., Y.S., K.H. and K.Y. wrote the manuscript. All authors have read and agreed to the published version of the manuscript.

**Funding:** This research received no external funding.

**Institutional Review Board Statement:** Ethical review and approval were waived for this study, due to the number of patients is less than three.

**Informed Consent Statement:** Informed consent was obtained from all subjects involved in the study.

**Conflicts of Interest:** The authors declare no conflict of interest.

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
