# Peer review of "Three Cases of Lymphocytic Infiltration of the Eyelid"

_dermatopathology, doi:10.3390/dermatopathology8020018_

Round 1
Reviewer 1 Report
The manuscript contains clinical and pathologic descriptions of three cases of patients diagnosed with LIS with a distinctive and unusual pattern strictly involving the upper eyelids. Two of the patients were biopsied within the first 4 months of presentation, while the third was biopsied later. The biopsy findings including immunohistochemistry detailing CD4 vs CD8 subsets in each of this patients can be of great interest to the readers, particularly with the temporal relationship of the findings that his not well-documented to my knowledge. The clinical images are excellent. I would like for the authors to also address why Morbihan's disease was excluded (i.e., that there was no history of rosacea, and that the absence of dermal edema, dilated vessels, and the density and cellular composition of the inflammation favors LIS. Grammar should be improved, and please see my specific comments below:
Lines 30-31: I would suggest: "The differential diagnosis includes. . . "
Lines 49 and 63: Please change "well" to "good or excellent".
Lines 47 and 62, and Figure 4 (d, h): It is stated in the figure legend and text that "CD8 is almost negative" but there appear to be significant numbers of CD8+ cells in the image. Although these are outnumbered by CD4+ cells, the number and density CD8+ cells in the Cases 1 and 2 are similar to, or exceed the number of CD8+ T-cells in Case 3. This can be easily rectified by qualifying the infiltrates in Cases 1 and 2 as having a predominance of CD4+ cells, with a significant minority of CD8+ lymphocytes.
Line 77: Change "were" to "was"
Line 79: Correct grammar, example: "The diagnosis of LIE of the face was rendered and . . . "
Line 114-115: Please clarify this statement. I am unclear about the mechanism of CD4+ T-cells promoting B-cell development in the skin (citation needed). CD20 was not reported in this series.
Line 91-92: I would also mention Morbihan's disease, and perhaps state whether any of the patients had a history of rosacea.
Line 125-126: Please edit this sentence, I do not understand the intent of the authors.
Author Response
Responses to the comments of Reviewer #1
- Lines 30-31: I would suggest: "The differential diagnosis includes. . . "
Response: We have changed to the description as you suggested.
- Lines 49 and 63: Please change "well" to "good or excellent".
Response: Thank you for your suggestion. We have changed the terms.
- Lines 47 and 62, and Figure 4 (d, h): It is stated in the figure legend and text that "CD8 is almost negative" but there appear to be significant numbers of CD8+ cells in the image. Although these are outnumbered by CD4+ cells, the number and density CD8+ cells in the Cases 1 and 2 are similar to, or exceed the number of CD8+ T-cells in Case 3. This can be easily rectified by qualifying the infiltrates in Cases 1 and 2 as having a predominance of CD4+ cells, with a significant minority of CD8+ lymphocytes.
Response: Thank you for the comment. As pointed out, small amount of CD8 + T cells were found in cases 1 and 2. We have changed the description as indicated case report section.
- Line 77: Change "were" to "was”
Response: The word has been changed.
- Line 79: Correct grammar, example: "The diagnosis of LIE of the face was rendered and . . . "
Response: Thank you for your suggestion. We have corrected the sentence as follow:
The diagnosis of LIE was made, and we started treatment with 8 mg/day of oral methylprednisolone.
- Line 114-115: Please clarify this statement. I am unclear about the mechanism of CD4+ T-cells promoting B-cell development in the skin (citation needed). CD20 was not reported in this series.
Response: We have performed the additional staining using CD20 antibody (Figure 4f,l,r). And the number of CD20+ B cells was few in the section for all three cases. We have deleted the sentences you pointed out. We apologizes for the confusion.
- Line 91-92: I would also mention Morbihan's disease, and perhaps state whether any of the patients had a history of rosacea.
Response: We thank the reviewer for the constructive comment. Morbihan’s disease should also be considered as a differential diagnosis. But in the current three cases, they didn’t have the history of acne or rosacea. And histopathological findings are different from those reported as Morbihan’s disease in the past. We reflected your great suggestion in our manuscript with some references.
- Line 125-126: Please edit this sentence, I do not understand the intent of the authors.
Response: Thank you for your suggestion. We thought LIE may be the a distinguished category of LIS. We have changed the description in the discussion.

Reviewer 2 Report
The manuscript of Sugioka et al. describes three cases of lymphocytic infiltration of the skin (LIS) involving the eyelid. Jessner LIS is a not well-characterized disease. The manuscript is well-structured and describes properly the patients included. However, in my opinion, the current form of the manuscript is not adding relevant information of the disease compared with papers already published. Further immunohistochemical markers and molecular studies could be performed to characterize better the skin infiltrates. Bibliographic review should be expanded; the format of the references should be uniform.
Author Response
Responses to the comments of Reviewer #2
- The manuscript of Sugioka et al. describes three cases of lymphocytic infiltration of the skin (LIS) involving the eyelid. Jessner LIS is a not well-characterized disease. The manuscript is well-structured and describes properly the patients included. However, in my opinion, the current form of the manuscript is not adding relevant information of the disease compared with papers already published. Further immunohistochemical markers and molecular studies could be performed to characterize better the skin infiltrates. Bibliographic review should be expanded; the format of the references should be uniform.
Response: Thank you for the comment. Although there have been reports of LIS in the past, the current cases we experienced are the circumscribed edematous indurative erythematous lesion localized to the eye, which we have succeeded in treating with low to middle dosage of oral steroids. Additional immunohistochemical staining and alcian blue staining have been supplemented and references have been added. The format of the references was arranged. Our manuscript has been proofread by native English speaker. We appreciated for your helpful comments.

Round 2
Reviewer 2 Report
The new version of the manuscript of Sugioka et al. improves the previous one regarding the quality of English, Bibliography and Discussion. However, the addition of CD20 staining and negativity of Alcian blue does not add relevant information. The information included in the Discussion about Morbihan disease adds information, but I think that 5 references are not necessary to support the added paragraph.
Jessner lymphocytic infiltration of the skin (JLIS) is a not well-characterized condition that histologically shows a lymphocytic infiltrate in the dermis. References 9 and 10 mention the current debate about whether this disease is a unique entity or part of a spectrum of other cutaneous diseases. Some authors consider JLIS as a variant of lupus (ref. 9). In my opinion, the authors should remark this controversy in the manuscript and give arguments favoring their own opinion. Moreover, as CD123 has been used to classify such diseases and the authors should include in the study CD123 immunostaining of the cases. I also suggest including follow-up data of the cases.
Author Response
Responses to the comments of Reviewer #2
The new version of the manuscript of Sugioka et al. improves the previous one regarding the quality of English, Bibliography and Discussion. However, the addition of CD20 staining and negativity of Alcian blue does not add relevant information. The information included in the Discussion about Morbihan disease adds information, but I think that 5 references are not necessary to support the added paragraph.
Jessner lymphocytic infiltration of the skin (JLIS) is a not well-characterized condition that histologically shows a lymphocytic infiltrate in the dermis. References 9 and 10 mention the current debate about whether this disease is a unique entity or part of a spectrum of other cutaneous diseases. Some authors consider JLIS as a variant of lupus (ref. 9). In my opinion, the authors should remark this controversy in the manuscript and give arguments favoring their own opinion. Moreover, as CD123 has been used to classify such diseases and the authors should include in the study CD123 immunostaining of the cases. I also suggest including follow-up data of the cases.
Response: Thank you for the informative and educational comment. According to your suggestion, we have edited and expanded the discussion section as typed in red. CD123 staining has been supplemented. Through the immune staining and histopathological character, we may suggest the proposal that lymphocytic infiltration of the eyelid is the distinct different category apart from JLIS. Please allow us to keep reference 3 to 7 to explain Morbihan disease. Again we really appreciated for your helpful comments.
